# HIV incidence and associated risk factors in adolescent girls and young women in South Africa: A population-based cohort study

Lara Lewis[1]*, Ayesha B. M. Kharsany[1,2], Hilton Humphries[1,3], Brendan Maughan-Brown[4], Sean Beckett[5], Kaymarlin Govender[5], Cherie Cawood[6], David Khanyile[6], Gavin George[5,7]

1 Centre for the AIDS Programme of Research in South Africa, University of KwaZulu–Natal, Durban, South Africa, 2 School of Laboratory Medicine and Medical Science, Nelson R Mandela School of Medicine, University of KwaZulu-Natal, Durban, South Africa, 3 Department of Psychology, School of Applied Human Sciences, University of KwaZulu-Natal, Durban, South Africa, 4 Southern Africa Labour and Development Research Unit, University of Cape Town, Cape Town, South Africa, 5 HIV Economics and AIDS Research Division, University of KwaZulu-Natal, Durban, South Africa, 6 Epicentre AIDs Risk Management, Cape Town, South Africa, 7 Division of Social Medicine and Global Health, Lund University, Lund, Sweden

* lara.lewis@caprisa.org

**Editor:** Hamid Sharifi, HIV/STI Surveillance Research Center and WHO Collaborating Center for HIV Surveillance, Institute for Future Studies in Health, Kerman University of Medical Sciences, ISLAMIC REPUBLIC OF IRAN

## Abstract

### Background

In sub-Saharan Africa, high HIV incidence rates in adolescent girls and young women (AGYW) persist despite extensive HIV prevention efforts.

### Methods

A prospective cohort of 2,710 HIV-negative AGYW (15–24 years) in KwaZulu-Natal, South Africa were interviewed at baseline and followed-up approximately 18 months later (2014–2017). Associations between HIV seroconversion and socio-demographic and behavioural variables measured at baseline and follow-up were examined using Cox regression and a proximate determinants framework. Inter-relationships between determinants were measured using logistic regression. Separate models were built for 15–19 and 20-24-year-olds.

### Results

Weighted HIV incidence was 3.92 per 100 person-years (95% confidence interval: 3.27–4.69; 163 seroconversions over 4,016 person-years). Among 15-19-year-olds, absence of family support (adjusted hazards ratio (aHR): 3.82 (1.89–7.72)), having a circumcised partner (aHR: 0.5 (0.27–0.94)) or one who was HIV-positive and not on antiretroviral therapy (ART) (aHR: 6.21 (2.56–15.06)) were associated with HIV incidence. Those reporting an absence of family support were also more likely to report >1 partner during follow-up (odds ratio (OR): 2.7(1.11–6.57)). Among 20-24-year-olds, failure to complete secondary school (aHR: 1.89 (1.11–3.21)), inconsistent condom use (aHR: 3.01 (1.14–7.96)) and reporting partner(s) who were HIV-positive and not on ART (aHR: 7.75 (3.06–19.66)) were associated with HIV incidence. Failure to complete secondary school among 20-24-year-

**Data Availability Statement:** All relevant data are within the paper and its Supporting information files.

**Funding:** The HIV Incidence Provincial Surveillance System (HIPSS) is funded by a cooperative agreement (3U2GGH000372) between Epicentre and the Centers for Disease Control and Prevention (CDC). ABMK is supported by a joint South Africa–U.S. Program for Collaborative Biomedical Research, National Institutes of Health grant (R01HD083343), the South African Department of Science and Innovation and the National Research Foundation's Centre of Excellence in HIV Prevention (Grant 96354). Support was provided to BMB by the National Research Foundation, South Africa, through the Research Career Advancement Fellowship. The content, findings and conclusions in this paper are those of the author(s) and do not necessarily represent the official views of the Centers for Disease Control and Prevention, or any other funder.

**Competing interests:** The authors have declared that no competing interests exist.

olds was associated with inconsistent condom use (OR: 1.82 (1.20–2.77)) and reporting an HIV-positive partner not on ART (OR: 3.53(1.59–7.82)) or an uncircumcised partner (OR: 1.39 (1.08–1.82).

## Conclusion

Absence of family support and incomplete schooling are associated with risky sexual behaviours and HIV acquisition in AGYW. In addition, partner-level prevention—condom use, medical circumcision, and viral suppression–continue to play an important role in reducing HIV risk in AGYW. These findings support the use of combination HIV prevention programs that consider structural as well as biological and behavioural HIV risk factors in their design.

## Introduction

Reducing HIV incidence rates among adolescent girls and young women aged 15–24 years (AGYW) in sub-Saharan Africa (SSA) is a key focus of the UNAIDS Global AIDS Strategy [1]. AGYW bear an inordinate burden of HIV risk; approximately 4,900 AGYW became infected with HIV every week in 2021 [2]. In South Africa, the country with the largest global burden of HIV, approximately a third of all new HIV infections are among AGYW [3]. Recent data suggests that HIV incidence rates in AGYW in this region have declined but the risk of HIV in this sub-group remains substantial [4–6]. A clear understanding of the socio-demographic, behavioural and biological determinants of HIV incidence in this vulnerable group is therefore critical for the development and success of global HIV prevention programmes.

The period between age 15 and 24 years is one of considerable biological, cognitive and social change for young women [7]. It marks the transition from child to adult, the move from school to employment, the initiation of sexual activity and, for many young women in South Africa, the beginning of motherhood. Previous research has highlighted several socioeconomic, behavioural and biological characteristics of AGYW that contribute to the disproportionately high HIV incidence rates experienced during this time period [8, 9]. Engagement in sexual relationships with men aged 5 or more years older is a key determinant of risk in AGYW, the primary reason being that, due to the aggregating nature of HIV prevalence in adult men, older male partners are more likely to be HIV-positive than younger ones [10, 11]. Previous research has also suggested that age-disparate relationships are associated with engagement in risky sexual behaviours such as inconsistent condom use and transactional sex [12]. AGYW are biologically more susceptible to HIV than young men owing to the comparatively larger surface area of the cervix-vagina mucosa, the longer HIV mucosal exposure time and differences in the mucosal immunology [13, 14]. High prevalence of other sexually transmitted infections (STIs) among AGYW in SSA also predispose them to a higher risk of HIV acquisition [15, 16]. In addition to behavioural and biological determinants, a number of socioeconomic factors have been associated with HIV incidence in AGYW, including but not limited to, incomplete schooling [17–21], orphanhood [22], financial insecurity [23] and gender inequalities [24].

While literature identifying possible risk factors for HIV infection in African women is extensive, few studies have had access to large population-based cohorts of AGYW in high-HIV epidemic settings. Most studies have relied on data collected either during cross-sectional surveys or during clinical trials for HIV prevention interventions on women of varying ages, making it difficult to establish causality and generalize results, respectively.

Additionally, many studies investigating the determinants of HIV risk in AGYW have been unsuccessful in delineating causal pathways of HIV acquisition and exploring the relationship between socioeconomic and behavioural/biological drivers of HIV risk. In this study, we identify determinants of HIV incidence in a group of AGYW in a hyperendemic setting using data from a large population-based, cohort study and a proximate determinants framework [25] to structure the analyses. To better understand the pathways through which HIV infections may occur, we measured associations between underlying social/demographic factors and behavioural factors that directly affect the likelihood of an individual being exposed and susceptible to HIV.

## Materials and methods

### Study design and setting

The study was undertaken in KwaZulu-Natal, South Africa, a province with an estimated HIV prevalence of 27% among individuals aged 15–49 [3]. Contraceptive services, HIV testing and treatment, voluntary medical male circumcision and provision of HIV preexposure and post-exposure prophylaxis are freely available through primary health care clinics.

We analysed data from the HIV Incidence Provincial Surveillance System (HIPSS) conducted in the uMgungundlovu district of KwaZulu-Natal [26]. HIPSS comprised of two serial cross-sectional household surveys, with two embedded HIV-negative cohorts comprising of a single follow-up visit. The first survey was conducted between June 2014 and June 2015 and the second between June 2015 and June 2016, and the follow-up visits were completed by January 2017 and August 2017 respectively. Fingerprint biometrics were used to confirm the identity of eligible participants for the follow-up visit. Individuals could be included in both surveys and cohorts if selected, however the overlap was minimal [5].

Multistage sampling was used to select the sample. One individual per household was selected at random and enrolled in the survey on condition they were aged between 15 and 49 years and provided peripheral blood samples for laboratory HIV and pregnancy testing. For those enrolled in the first survey, STI testing for *Neisseria gonorrhoeae*, *Chlamydia trachomatis*, *Trichomonas vaginalis* and *Mycoplasma genitalium by* multiplex PCR and the serological detection of antibodies to syphilis and HSV-2 was also conducted [16]. Individuals were enrolled in the cohorts if they were HIV negative at survey enrolment and aged between 15 and 35 years. HIV testing was also conducted at the follow-up visit. Standardized face-to-face interviews in which participants answered a series of questions relating to their socio-demographic status and behaviour were performed at enrolment and follow-up. Behavioural data collected at follow-up related specifically to sexual behaviour that occurred in the 12 months preceding follow-up. Further details of the study have been previously published [5]. The present study used data collected at enrolment and follow-up on participants aged 15–24 years. Data collected in the two cohorts were combined and analysed as one successive cohort that began in June 2014 and ended in August 2017.

### Conceptual approach

A proximate determinants framework was used to structure the analysis [25]. Proximate determinants of HIV are behavioural and biological characteristics (of the individual and partner) that directly affect the likelihood of an individual being exposed and susceptible to HIV, for example condom use or number of sex partners. Underlying determinants are defined as social, economic, and demographic factors that operate through proximate determinants to influence the likelihood of being exposed to HIV, for example, marital status

or education. Three sets of analyses were conducted to quantify the association between proximate/underlying determinants and HIV incidence and to quantify the association between proximate and underlying determinants themselves:

1. The association between proximate determinants and HIV incidence was quantified.

2. The association between underlying determinants and HIV incidence was measured. This association was first measured without adjusting for proximate determinants. Thereafter, the association between underlying determinants and HIV incidence was measured after adjusting for proximate determinants as well. However, if the proximate determinants acted as mediators between underlying determinants and HIV infection, we expect that a model which already includes proximate determinants would not be influenced by the inclusion of underlying determinants.

3. Finally, to explore how underlying determinants may influence proximate determinants of HIV incidence, the association between the underlying and proximate determinants was quantified.

Analyses were performed separately for 15-19-year-olds and 20-24-year olds as we hypothesized that the factors affecting young girls who are still of school going age are likely to be different from those affecting women who are out of secondary-school, possibly in tertiary education or seeking employment.

## Variables

**Outcome measure.**   The main outcome measure in the study was HIV incidence. As the date of each HIV seroconversion was unknown, it was estimated to be the mid-point between the enrolment date and follow-up visit date.

Underlying and proximate determinants comprised socio-demographic, behavioural and biological factors that were measured in the HIPSS survey and ones that have been determined as influencing HIV incidence in literature [19, 27, 28].

**Underlying determinants.**   All underlying determinants were based on variables measured at enrolment as they were expected to be time invariant or relatively stable over the follow-up period. Variables identified as potential underlying determinants included participant age, secondary school completion, total household income, urban/rural residence, orphan status (defined as being both maternally and paternally orphaned) and report of *any* family (emotional or financial, in the form of money, food, education or shelter) support in the preceding 12 months. Since the normal school-completion age is approximately 18 years in South Africa, the education variable could only be meaningfully interpreted as a measure of risk in AGYW aged 20–24 years.

**Proximate determinants.**   Proximate determinants included the number of lifetime sexual partners reported at enrolment and the number of partners during follow-up, consistent condom use during follow-up, number of STIs at enrolment, engagement in transactional sex during follow-up, and partner characteristics namely, partner(s) age difference (at least one partner $\geq$ 5 years older versus not), partner(s) circumcision status (all partners circumcised at the beginning or during relationship, versus not), and partner(s) HIV status (at least one sexual partner HIV-positive and not on ART versus none). Partner characteristics were measured using data on the most recent partner reported at enrolment and partnerships occurring during the 12 months preceding follow-up. The circumcision variable did not differentiate between medical or traditional circumcision however prevalence of traditional circumcision in this area is low [29].

## Statistical analysis

The association between identified underlying and proximate determinants and HIV incidence was estimated using Cox proportional hazards models. Since data on orphan status and STI testing data was only collected for one of the two cohorts, orphan and STI status were excluded from the Cox regression although included in descriptive analysis. Univariable regression was first performed followed by multivariable regression. All variables included in the univariable were included in the multivariable regression as all variables were hypothesized to be associated with HIV incidence. The association between the underlying determinants and proximate determinants found to be significantly associated with HIV incidence was measured using logistic regression. Unless otherwise stated, all AGYW, regardless of whether they reported having sex before study enrolment, were included in the analysis. However, models that incorporated sexual behaviour variables that were based on follow-up data excluded, by necessity, AGYW who reported not being sexually active in the 12 months preceding follow-up. In addition, 32 AGYW who reported having had sex at enrolment but reported never having sex at follow-up were excluded from analyses using measures of sexual behaviour. Analyses were conducted in SAS, version 9.4 (SAS Institute Inc). Survey weights, which accounted for the unequal probability of selection of each individual and adjusted for differences in non-response rates across age and gender groups, and a significance level of 0.05 were used in analysis.

## Ethical approval

The HIPSS study protocol, informed consent and data collections forms were reviewed and approved by the University of KwaZulu-Natal Biomedical Research Ethics Committee (BF269/13), KwaZulu-Natal Provincial Department of Health (HRKM 08/14) and the Associate Director of Science of the Centre for Global Health (CGH) at the United States Centre for Disease Control and Prevention (CDC) in Atlanta, United States of America (CGH 2014–080).

Written informed consent was obtained from participants 18 years and older and parental/guardian /caregiver consent for participants 15 to <18 years of age and individual assent from participants 15 to <18 years of age for study participation. A separate written informed consent was obtained for long term sample storage for confirmation of any discrepant or uncertain results and for future testing if indicated. Each participant was assigned a unique study participant identification number so that their personal data or laboratory results could not be linked to any personal identifiers such as name.

## Results

Of the 3518 AGYW who were HIV-negative at the time of enrolment, 2710 (77.0%) completed a follow-visit with a median follow-up time of 17 [(Interquartile range (IQR) 15–21] months. A total of 163 of the 2710 AGYW seroconverted during follow-up resulting in an incidence rate of 3.92 (95% CI: 3.27–4.69) per 100 person-years, and an incidence rate for AGYW aged 15–19 and 20–24 years of 3.74 (95% CI: 2.87–4.86) and 4.13 (95% CI: 3.20–5.33) per 100 person-years respectively (Table 1).

HIV incidence rates for various sub-groups of AGYW are reported in Table 1. Among those who reported never having had sex at enrolment, HIV incidence was 1.71 (95% CI: 1.12–2.60) per 100 person-years. Among AGYW below the age of 18, 11.8% reported being orphans, with an estimated incidence rate of 9.23 (95% CI: 3.86–22.17) per 100 person-years. Most AGYW aged 15–19 years (91.9%) indicated receiving emotional, informational and/or financial support from at least one family member in the 12 months leading up to enrolment in the study. Among those that did not receive this support, HIV incidence was 10.40 (95% CI:

**Table 1.** Baseline and follow-up characteristics and HIV incidence in adolescent girls and young women enrolled in the HIPSS cohort.

| Participant characteristics | %(n) | | | HIV incidence rate (#seroconversions/person-years) | | |
|---|---|---|---|---|---|---|
| | 15–19 years | 20–24 years | Total | 15–19 years | 20–24 years | Total |
| Overall | 100(1403) | 100(1307) | 100(2710) | 3.74(92/2071) | 4.13(71/1945) | 3.92(163/4016) |
| Highest education level at enrolment (≥ 20 years) | | | | | | |
| Did not complete secondary school | n/a | 31.8(410) | n/a | n/a | 5.94 (35/602) | n/a |
| Completed secondary or tertiary schooling | n/a | 68.2(897) | n/a | n/a | 3.31 (47/1343) | n/a |
| Total household income at enrolment | | | | | | |
| R0—R500 pm | 8.7(129) | 11.2(171) | 9.8(300) | 2.53(8/209) | 2.96(12/278) | 2.76(20/487) |
| R501—R2,500 pm | 44.5(636) | 44.1(577) | 44.3(1213) | 4.72(41/934) | 5.09(44/851) | 4.89(85/1786) |
| R2,501—R6,000 pm | 36.4(442) | 33.6(373) | 35.1(815) | 3.69(24/626) | 2.22(14/533) | 3.04(38/1160) |
| greater than R6,000 pm | 10.5(137) | 11.2(124) | 10.8(261) | 1.89(6/198) | 5.06(7/173) | 3.38(13/372) |
| Missing | 59 | 62 | 121 | | | |
| Location of residence at enrolment | | | | | | |
| Urban | 49.3(864) | 47.0(791) | 48.3(1655) | 3.5(50/1279) | 4.23(50/1189) | 3.83(100/2468) |
| Rural | 50.7(539) | 53.0(516) | 51.7(1055) | 3.97(31/791) | 4.04(32/756) | 4.00(63/1547) |
| Orphan status at enrolment (<18 years)[a] | | | | | | |
| Either father or mother or both are alive | 88.2(346) | n/a | n/a | 2.13(11/479) | n/a | n/a |
| Both mother and father are deceased | 11.8(43) | n/a | n/a | 9.23(5/59) | n/a | n/a |
| Missing | 25 | | | | | |
| Family support at enrolment | | | | | | |
| Receives emotional/financial support from family | 91.9(1270) | 86.1(1084) | 89.2(2354) | 3.17(66/1867) | 4.15(66/1594) | 3.6(132/3461) |
| Receives no emotional/financial support from family | 8.1(133) | 13.9(223) | 10.8(356) | 10.4(15/203) | 4.02(16/351) | 6.46(31/554) |
| Number of lifetime sexual partners at enrolment | | | | | | |
| None | 58.8(787) | 14.4(167) | 38.2(954) | 1.77(24/1185) | 1.39(5/253) | 1.71(29/1439) |
| 1 partner | 25.5(369) | 37.6(472) | 31.1(841) | 4.6(25/537) | 4.03(23/711) | 4.28(48/1249) |
| 2–4 partners | 14.2(189) | 43.2(550) | 27.6(739) | 7.88(22/265) | 4.62(43/795) | 5.5(65/1061) |
| 5 or more partners | 1.5(24) | 4.9(75) | 3.1(99) | 11.32(5/32) | 4.92(6/108) | 6.61(11/141) |
| Missing | 34 | 43 | 77 | | | |
| Number of sexual partners in 12 months preceding follow-up[c] | | | | | | |
| None | 45.9(618) | 20.3(234) | 34(852) | 1.84(17/908) | 2.90(13/334) | 2.13(30/1243) |
| 1 partner | 50.3(728) | 75.1(995) | 61.8(1723) | 5.17(58/1076) | 4.12(59/1493) | 4.58(117/2569) |
| > 1 partner | 3.8(57) | 4.6(76) | 4.2(133) | 7.76(6/85) | 9.67(10/116) | 8.75(16/201) |
| Missing | 0 | 2 | 2 | | | |
| Partner(s) age[b] | | | | | | |
| At least one partner 5 or more years older | 36.2(321) | 40.7(465) | 41.3(786) | 5.97(34/473) | 3.29(27/705) | 4.37(61/1178) |
| No partners reported to be 5 or more years older | 63.8(528) | 59.3(709) | 63.9(1237) | 5.45(37/801) | 5.08(50/1060) | 5.25(87/1862) |
| Missing | 63 | 46 | 109 | | | |
| Partner(s) circumcision status[b] | | | | | | |
| All sexual partners reported to be circumcised | 66.4(517) | 55.7(630) | 60.2(1147) | 3.63(32/770) | 4.88(41/931) | 4.29(73/1702) |
| At least one sexual partner was reported not to be circumcised | 33.6(277) | 44.3(505) | 39.8(782) | 7.25(29/407) | 3.79(34/760) | 5.00(63/1167) |
| Missing | 118 | 85 | 203 | | | |
| Partner(s) HIV status[b] | | | | | | |
| No sexual partner reported to be HIV positive and not on ART | 98.3(832) | 97.9(1143) | 98.1(1975) | 5.09(65/1229) | 4.10(70/1710) | 4.53(135/2939) |
| At least one sexual partner reported to be HIV positive and not on ART | 1.7(17) | 2.1(31) | 1.9(48) | 30.39(5/22) | 18.44(7/42) | 22.77(12/64) |
| Missing | 63 | 46 | 109 | | | |
| Condom use[c] | | | | | | |
| Reported always using condoms during sex | 23.2(188) | 19.6(205) | 21.2(393) | 5.61(15/282) | 2.02(8/309) | 3.73(23/592) |

*(Continued)*

**Table 1.** (Continued)

| Participant characteristics | %(n) | | | HIV incidence rate (#seroconversions/person-years) | | |
|---|---|---|---|---|---|---|
| | 15–19 years | 20–24 years | Total | 15–19 years | 20–24 years | Total |
| Reported not always using condom during sex | 76.8(593) | 80.4(858) | 78.8(1451) | 5.3(49/873) | 5.07(61/1289) | 5.17(110/2162) |
| Missing | 4 | 8 | 12 | | | |
| Transactional sex[c] | | | | | | |
| Reported receiving money/gifts for sex | 9.6(74) | 9.2(106) | 9.3(180) | 5.86(7/109) | 6.14(7/154) | 6.01(14/264) |
| Reported never receiving money/gifts for sex | 90.4(711) | 90.8(965) | 90.7(1676) | 5.29(57/1051) | 4.26(62/1457) | 4.71(119/2509) |
| Pregnancy history at enrolment | | | | | | |
| Currently or previously pregnant | 23.4(354) | 64.1(851) | 42.3(1205) | 5.41(31/509) | 4.42(59/1257) | 4.71(90/1766) |
| Never pregnant | 76.6(1042) | 35.9(448) | 57.7(1490) | 3.27(50/1548) | 3.68(23/673) | 3.39(73/2222) |
| Missing | 7 | 8 | 15 | | | |
| Contraception at enrolment | | | | | | |
| Currently on a contraceptive method | 27.1(391) | 63.7(823) | 44.1(1214) | 5.98(33/562) | 4.48(54/1211) | 4.97(87/1773) |
| Not currently on a contraceptive method | 72.9(1012) | 36.3(484) | 55.9(1496) | 2.93(48/1508) | 3.52(28/734) | 3.11(76/2242) |
| Reported use of PrEP[c] | 1.3 (17) | 2.1 (28) | 1.7 (45) | 2.09(1/28) | 0(0/50) | 0.83(1/78) |
| STI at enrolment[d] | | | | | | |
| No STIs detected | 62.4 (393) | 39.1 (254) | 51.8 (647) | 2.91(16/626) | 2.54(13/405) | 2.79(29/1031) |
| One STI detected | 25.6 (187) | 40.7 (257) | 32.5 (444) | 5.71(20/292) | 3.74(15/411) | 4.59(35/704) |
| More than one STI detected | 12.0 (80) | 20.1 (128) | 15.7 (208) | 11.72(15/119) | 7.32(13/205) | 9.1(28/325) |

[a]Orphan status only available for second cohort of HIPSS.

[b]Partner data based on data on most recent partner at enrolment and data from partner(s) reported in the 12 months preceding follow-up.

[c]Based on behavioural data reported for the 12 months preceding cohort follow-up.

[d]Neisseria gonorrhoeae, Chlamydia trachomatis, Trichomonas vaginalis, Mycoplasma genitalium, syphilis and HSV-2. STI testing was only conducted for participants in the first HIPSS cohort.

5.67–19.09) per 100 person-years compared to 3.17 (95% CI: 2.38–4.21) among those who did. Almost a third of AGYW aged 20-24-year-olds had not yet completed secondary school although their age exceeded the age-norm of secondary schools. Among them, the HIV incidence rate was 5.94 (95% CI: 3.96–8.89) per 100 person-years. Just under 2% (n = 48) of all AGYW reported knowingly having had sex with an individual who was HIV positive and not on ART and of these AGYW, 25% (n = 12) seroconverted during follow-up.

Compared to those with follow-up data, HIV-negative AGYW who were not available for follow-up (n = 808, 23.0%) were more likely to be aged between 20 and 24 (48.2% versus 54.7%) and reside in a rural area (51.7% versus 76.6%). However, the two groups were comparable with respect to education levels, household income, number of lifetime sexual partnerships and STI status at baseline.

## Association between proximate determinants and HIV

The results from bivariable and multivariable models predicting the association between proximate determinants and HIV incidence are shown in Table 2. AGYW who did not report having sex in the 12 months preceding follow-up (n = 852) were excluded from the multivariable analysis as they did not have sexual behaviour data. Among AGYW aged 15–19 years, an increase in the number of lifetime partners reported at enrolment was associated with an increase in HIV incidence (adjusted hazard ratio (aHR): 1.15 (95% CI: 1.01–1.31)). Reporting having a circumcised partner was protective against HIV (aHR = 0.5 (95% CI: 0.27–0.94)) and reporting one or more partners who were thought or known to be HIV positive and not on

**Table 2. Association between proximate determinants and HIV incidence in adolescent girls and young women enrolled in the HIPSS cohort.**

| Proximate determinants | 15-19-year-old women | | 20-24-year-old women | |
|---|---|---|---|---|
| | HR (95%CI) | HR adjusted for all proximate determinants (95%CI)[b] | HR (95%CI) | HR adjusted for all proximate determinants (95%CI)[b] |
| # Lifetime sexual partners at enrolment | 1.26(1.14–1.4)[a] | 1.15(1.01–1.31)[a] | 1.08(0.99–1.19) | 0.97(0.86–1.1) |
| # Sexual partners in 12 months preceding follow-up | | | | |
| 0 vs 1 | 1.16(0.55–2.46) | n/a | 1.07(0.51–2.27) | n/a |
| 2 or more vs 1 | 1.5(0.55–4.11) | 1.5(0.5–4.49) | 2.39(1.16–4.95)[a] | 2.85(1.36–5.97)[a] |
| Age of oldest partner[c] | 1.06(1–1.11)[a] | 1.03(0.96–1.1) | 0.97(0.9–1.05) | 0.98(0.92–1.05) |
| Partner(s) HIV status[c] | | | | |
| At least one partner known to be HIV positive and not on ART vs No partner reported to be HIV positive and not on ART | 5.91(2.57–13.59)[a] | 6.21(2.56–15.06)[a] | 4.49(1.86–10.84)[a] | 7.75(3.06–19.66)[a] |
| Partner(s) circumcision status[c] | | | | |
| All partners reported to be circumcised vs not all partners reported to be circumcised | 0.51(0.29–0.9)[a] | 0.5(0.27–0.94)[a] | 1.26(0.72–2.19) | 1.54(0.89–2.67) |
| Condom use in 12 months preceding follow-up[b] | | | | |
| Engaged in condomless sex vs did not report condomless sex | 0.94(0.47–1.9) | 0.98(0.46–2.12) | 2.5(1.05–5.93)[a] | 3.01(1.14–7.96)[a] |
| Transactional sex in 12 months preceding follow-up[b] | | | | |
| Engaged in transactional sex vs did not report transactional sex | 1.13(0.43–2.96) | 1.24(0.49–3.15) | 1.39(0.53–3.68) | 1.56(0.6–4.07) |

[a]Significant at a 5% significance level.

[b]For a sample of women who had sex in the 12 months preceding follow-up.

[c]For a sample of women with data on the most recent partner at enrolment and/or partnership data in the 12 months preceding follow-up.

ART was associated with higher HIV risk (aHR = 6.21 (95% CI: 2.56–15.06)). In bivariable models, an increase in the age of the AGYW's partner was associated with an increase in HIV incidence (HR = 1.06 (95% CI: 1–1.11)). However, this effect was not significant after controlling for other proximate determinants.

Among AGYW aged 20–24 years, those who reported 2 or more partners in the 12 months preceding their follow-up visit were at higher risk of acquiring HIV than those who only reported having 1 partner over that period (aHR: 2.85 (95% CI: 1.36–5.97)). HIV incidence was also higher among those who reported inconsistent condom use (aHR = 3.01 (95% CI: 1.14–7.96)) and having had sex with a partner who was thought or known to be HIV positive and not on ART (aHR = 7.75 (95% CI: 3.06–19.66)).

## Association between underlying determinants and HIV

The results from bivariable models and multivariable models (first adjusting for other underlying determinants and subsequently adjusting for other underlying determinants as well as proximate determinants) measuring the association between underlying determinants and HIV incidence are presented in Table 3. In the bivariate model and the model adjusting for other underlying determinants only, AGYW aged 15–19 years who did not receive support in the form of money or information or emotional support from family members in the 12 months preceding enrolment had higher HIV incidence (Ahr = 3.82

**Table 3. Association between underlying determinants and HIV incidence in adolescent girls and young women enrolled in the HIPSS cohort.**

| Underlying determinants | 15-19-year-old women | | | 20-24-year-old women | | |
|---|---|---|---|---|---|---|
| | HR (95% CI) | HR adjusted for all underlying determinants (95%CI) | HR adjusted for all underlying & proximate determinants (95%CI) | HR (95% CI) | HR adjusted for all underlying determinants (95%CI) | HR adjusted for all underlying & proximate determinants (95%CI) |
| Age at enrolment | 1.25 (1.08–1.46)[a] | 1.24 (1.06–1.45)[a] | 0.94(0.74–1.2) | 0.96 (0.82–1.13) | 0.92 (0.78–1.09) | 0.95(0.78–1.17) |
| Highest education at enrolment (≥ 20) | | | | | | |
| Did not complete secondary school vs completed secondary | n/a | n/a | n/a | 1.86 (1.07–3.24)[a] | 1.89 (1.11–3.21)[a] | 1.67(0.92–3.04) |
| Location of residence at enrolment | | | | | | |
| Urban vs rural | 0.85 (0.49–1.45) | 0.8 (0.48–1.34) | 1.14(0.6–2.15) | 0.98 (0.59–1.64) | 1.01 (0.6–1.69) | 0.92(0.51–1.65) |
| Household income at enrolment | | | | | | |
| R0 –R500 pm vs >R6000 pm | 1.38 (0.42–4.54) | 0.88 (0.25–3.09) | 1.41(0.31–6.43) | 0.58 (0.2–1.7) | 0.55 (0.19–1.59) | 0.43(0.1–1.83) |
| R501 –R2,500 pm vs >R6000 pm | 2.2 (0.84–5.75) | 2.2 (0.88–5.51) | 2.22(0.7–7.08) | 0.96 (0.37–2.53) | 0.89 (0.35–2.26) | 1.16(0.34–3.99) |
| R2,501 –R6,000 pm vs >R6000 pm | 1.77 (0.65–4.82) | 1.88 (0.71–4.95) | 1.23(0.35–4.27) | 0.42 (0.14–1.25) | 0.4 (0.14–1.2) | 0.56(0.13–2.32) |
| Family support at enrolment | | | | | | |
| Receives no family financial/emotional support vs receives family support | 3.29 (1.57–6.89)[a] | 3.82 (1.89–7.72)[a] | 1.97(0.76–5.11) | 1(0.55–1.83) | 1.11 (0.6–2.06) | 1.09(0.53–2.26) |

[a]Significant at a 5% significance level.

(95% CI: 1.89–7.72)). After including proximate determinants in the model, the effect of family support on HIV incidence was no longer statistically significant. Among AGYW aged 20–24 years, failure to complete secondary education was positively associated with HIV incidence after controlling for other underlying determinants (Ahr = 1.89 (95% CI: 1.11–3.21)). However, this association was only weakly significant after additionally adjusting for proximate determinants (Ahr = 1.67 (95% CI: 0.92–3.04)).

## Association between underlying and proximate determinants

Proximate determinants found to be associated with HIV incidence included number of partners during follow-up, partner(s) reported HIV and ART status, partner circumcision status and condom use. The association between underlying determinants found to be associated with HIV acquisition–education and family support–and these proximate determinants are presented in Table 4. In bivariable analyses, AGYW aged 20–24 years who had not completed their secondary education were found to be at greater risk of having an HIV-positive partner not on ART (odds ratio (OR) = 3.53 (95% CI: 1.59–7.82)) and engaging in condomless sex (OR = 1.82 (95% CI: 1.2–2.77)) and were less likely to have circumcised partners (OR = 0.72 (95% CI: 0.55–0.93)). Among AGYW aged 15–19 years who had engaged in sex in the 12 months preceding follow-up, those who lacked family support were more likely to have more than 1 sexual partner (OR = 2.7 (95% CI: 1.11–6.57)).

**Table 4. Association between underlying and proximate determinants of HIV in adolescent girls and young women enrolled in the HIPSS cohort.**

| | | Odds Ratio (95%CI) | | | |
|---|---|---|---|---|---|
| | | Proximate determinants | | | |
| | Underlying determinants | # Partners 12 months before follow-up (2 or more vs 1) | Partner(s) HIV status (at least one positive and not on ART vs not) | Partner(s) circumcision status (all circumcised vs not) | Condom use (inconsistent or no condom use vs consistent condom use) |
| **15–19 years** | Family support Receives no financial/emotional support from family, vs receives some support from family | 2.7(1.11–6.57)[a] | 2.25(0.64–7.88) | 1.47(0.91–2.37) | 1.88(0.88–3.98) |
| **20–24 years** | Highest education level (>20 years) Did not complete secondary school, vs completed secondary/tertiary | 1.62(0.92–2.85) | 3.53(1.59–7.82)[a] | 0.72(0.55–0.93)[a] | 1.82(1.2–2.77)[a] |

[a]Significant at a 5% significance level.

## Discussion

This study examined inter-relationships between socio-demographic, behavioural and biological variables and their association with HIV incidence in a large cohort of AGYW in a hyper-endemic area of South Africa. High HIV incidence rates were observed in all sub-groups of AGYW in this area, even in those in their first year after sexual debut. Proximate determinants found to be strongly positively associated with HIV incidence were the number of sexual partners at enrolment and follow-up, engagement in sex with partners who were uncircumcised, inconsistent condom use and having a sexual partner who was HIV positive and not on ART. Two underlying determinants—namely family support and secondary school completion—were found to be protective against HIV acquisition. The existence of family support and secondary school completion were also negatively associated with the probability of condomless sex, having an uncircumcised partner, having an HIV-positive partner not on ART and the number of sexual partners during follow-up, suggesting possible mechanisms through which these underlying determinants influenced HIV risk.

Approximately 1 in 3 AGYW aged 20 to 24 years in this region reported not completing secondary education. Their risk of acquiring HIV during follow-up was 89% higher than those that had completed their secondary education. AGYW with incomplete schooling were also found to be at increased risk of engaging in high-risk sexual behaviours, namely having sex with an HIV positive partner not on ART, having an uncircumcised partner and engaging in condomless sex, suggesting possible mechanisms through which they are exposed to higher risk of HIV. These findings are consistent with previous literature which has shown that increases in school attendance in AGYW is associated with a reduction in risky sexual behaviour [17, 20] and HIV incidence [18, 30]. Additional years of schooling likely provide young women with greater knowledge of HIV transmission and protection, which in turn reduce their willingness to engage in high-risk sexual behaviours. Through its positive impact on employment opportunities, education may also provide women with a higher level of economic independence from their partners, consequently allowing women to be more discerning about their choice of partner and decisions regarding their sexual health.

Among AGYW aged 15 to 19 years, family support emerged as an important protective factor against HIV acquisition. For those who reported having no emotional or financial support from family members in the 12 months preceding enrolment, HIV incidence rates were more than 3-fold higher during follow-up. Similarly, HIV incidence rates of double orphans were more than 4-fold that of those who had at least one parent alive. There are several ways in

which family support may reduce adolescents' likelihood of engaging in risky sexual behaviour and their HIV risk [31]. Communication and education regarding sexuality within families, parental monitoring and accountability to parents, and family support for academic endeavours are some of the aspects of family structures that assist in minimising sexual risk-taking among adolescents. AGYW who lack family support may also look to partnerships as a source of emotional and/or financial support, particularly adolescent young women for whom formal employment is difficult. Young women lacking family support may therefore be less discriminating about their choice of partner(s), and notably AGYW in this cohort who reported lacking family support had more partners during follow-up. These women may arguably also have less power in their relationships owing to an increased dependency on their partner(s). While evidence quantifying the effect of family support on HIV incidence in AGYW is limited, there is a growing body of evidence suggesting that programmes enhancing family support significantly reduce outcomes proximal to HIV infection [32, 33].

Partner characteristics were shown to have a significant impact on the HIV risk of AGYW. HIV incidence was approximately halved for those who exclusively engaged in sex with circumcised partners (AGYW aged 15–19 years) and for those who consistently used a condom during sex (AGYW aged 20–24 years). AGYW who reported having sex with partners who were thought or known to be HIV positive and not on ART (n = 48, 1.9%) had HIV incidence rates 5-fold higher than those that did not. Lastly, partner age was positively associated with HIV incidence among 15–19 year olds in bivariable analysis. This effect became statistically insignificant after controlling for variables capturing behaviours typically associated with age-disparate relationships–inconsistent condom use, engagement in transactional sex and low rates of partner viral suppression—suggesting that the model partially captured the mechanisms through which age-disparate partnerships increase HIV risk in AGYW. These findings highlight the importance of promoting couples HIV testing and counselling, continuing the roll-out of voluntary medical male circumcision programmes and of engaging men in HIV prevention services.

The results from this study suggest that a myriad of factors—structural, behavioural and biological—contribute to the persistently high HIV incidence rates among AGYW in this area. This underscores the need for the design and implementation of combination HIV prevention programs that address this wide spectrum of risks. The DREAMS program, which aims to reduce HIV incidence among AGYW in high-burden settings and to ensure that AGYW live Determined, Resilient, Empowered, AIDS-free, Mentored and Safe lives ("DREAMS"), is currently implemented in 16 countries in SSA and uses an evidence-based multi-sectoral strategy to address the multidimensional vulnerabilities of AGYW in this region [34]. While it is anticipated that the impact of such a complex program may only be seen in the long-term, promising signs of reductions in HIV incidence are already being observed [35].

These results should be interpreted in the context of the study limitations. The analysis was limited to data collected during the HIPSS study between 2014 and 2017 and excluded potential predictors of HIV incidence. In addition, educational achievement–a key covariate used in multivariable models—may be a proxy for other covariates such as personality traits that could not be controlled for in the models. Social desirability bias may have led to the underreporting of risky sexual behaviour. In addition, although most participants were followed up 18 months after enrolment, behavioural questions at follow-up captured sexual behaviour in the preceding 12 months leaving a gap in behavioural data of approximately 6 months. To partially mitigate this, the behavioural variables used in analysis incorporated both baseline and follow-up data. However, any relationships starting and ending in the period immediately after enrolment and approximately 6 months after may have been excluded. Although less than 5% of AGYW reported having more than 2 partners in the 12 months preceding follow-up, most of

these women only provided data on 1 of these partners. Missing data can lead to a loss of statistical power or introduce an unexpected selection bias. However, in this study the number of missing responses to questions in the baseline survey and cohort follow-up visit were generally low, and a comparison of characteristics of those lost to follow-up to those who were followed did not suggest that the two groups were inherently different.

## Conclusions

This study highlights the endemic levels of HIV incidence among AGYW living in this region. It supports findings from previous literature that showed that retaining AGYW in school and encouraging partner-level prevention through voluntary medical male circumcision, condom use and HIV treatment as prevention, are critical to preventing new infections among AGYW. Additionally, it underscores the importance of family support structures in reducing the likelihood of AGYW engaging in risky sexual behaviour and acquiring HIV. The challenge remains to design and implement community-based HIV prevention programs that effectively address these issues.

## Supporting information

**S1 File. Sample dataset.**
(PDF)

## Acknowledgments

We thank all the study participants, study staff, co-investigators from Epicentre, CAPRISA, HEARD, NICD and CDC and district primary health care clinic staff. We thank our collaborating partners: The National Department of Health, Provincial KwaZulu-Natal Department of Health, uMgungundlovu Health District, the uMgungundlovu District AIDS Council, HIV and AIDS / STI / TB (HAST) unit KwaZulu-Natal, local municipal and traditional leaders, and community members for all their support throughout the study.

## Author Contributions

**Conceptualization:** Lara Lewis, Ayesha B. M. Kharsany.

**Data curation:** Lara Lewis.

**Formal analysis:** Lara Lewis.

**Methodology:** Lara Lewis.

**Validation:** Lara Lewis.

**Writing – original draft:** Lara Lewis.

**Writing – review & editing:** Ayesha B. M. Kharsany, Hilton Humphries, Brendan Maughan-Brown, Sean Beckett, Kaymarlin Govender, Cherie Cawood, David Khanyile, Gavin George.

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
