## [Decision Letter · Decision Letter 0]

1 Aug 2022

PONE-D-22-19202HIV incidence and associated risk factors in adolescent girls and young women in South Africa: A population-based cohort studyPLOS ONE

Dear Mrs. Lara Lewis

Thank you for submitting your manuscript to PLOS ONE. After careful consideration, we feel that it has merit but does not fully meet PLOS ONE’s publication criteria as it currently stands. Therefore, we invite you to submit a revised version of the manuscript that addresses the points raised during the review process. Please submit your revised manuscript by Sep 15 2022 11:59PM. If you will need more time than this to complete your revisions, please reply to this message or contact the journal office at plosone@plos.org. Please include the following items when submitting your revised manuscript:A rebuttal letter that responds to each point raised by the academic editor and reviewer(s). You should upload this letter as a separate file labeled 'Response to Reviewers'.A marked-up copy of your manuscript that highlights changes made to the original version. You should upload this as a separate file labeled 'Revised Manuscript with Track Changes'.An unmarked version of your revised paper without tracked changes. You should upload this as a separate file labeled 'Manuscript'.

We look forward to receiving your revised manuscript.

Kind regards,

Hamid Sharifi

Academic Editor

PLOS ONE

Journal Requirements:

3. In the ethics statement in the Methods and online submission information, please ensure that you have specified (1) whether consent was informed and (2) what type you obtained (for instance, written or verbal, and if verbal, how it was documented and witnessed). If your study included minors, state whether you obtained consent from parents or guardians. If the need for consent was waived by the ethics committee, please include this information.

Reviewers' comments:

Reviewer's Responses to Questions

**Comments to the Author**

1. Is the manuscript technically sound, and do the data support the conclusions?

Reviewer #1: Partly

Reviewer #2: Yes

Reviewer #3: Yes

2. Has the statistical analysis been performed appropriately and rigorously? 

Reviewer #1: I Don't Know

Reviewer #2: Yes

Reviewer #3: Yes

3. Have the authors made all data underlying the findings in their manuscript fully available?

Reviewer #1: Yes

Reviewer #2: Yes

Reviewer #3: Yes

4. Is the manuscript presented in an intelligible fashion and written in standard English?

Reviewer #1: Yes

Reviewer #2: Yes

Reviewer #3: Yes

5. Review Comments to the Author

Reviewer #1: This is a cohort study aiming to identify determinants of HIV incidence in adolescent girls and young women in a hyperendemic setting in South Africa. It is a very interesting study; however, it presents some gaps, especially in the methodology, and must be improved.

1. I missed the sample size estimation in the Methods section. Is your sample size statistically significant, and the findings of this study can be generalized?

2. The inclusion and exclusion criteria are not clearly stated. They should be presented in the Study design and setting section.

3. An important piece of information not available in the paper is how the two cohorts were linked.

4. How did you analyze the associated factors by statistical or theoretical criteria? In fact, the authors didn’t focus on how this analysis was done in the Statistical analysis section, although this is one of the paper's objectives.

5. Why did you use logistic regression to estimate the association between the underlying and proximate determinants significantly associated with HIV incidence?

6. Role of the Funder/Sponsor: This section can be stated at the end of the paper, after the conclusions.

7. Association between underlying and proximate determinants: This analysis is very confusing! I think a hierarchical analysis model would be better in this context, splitting by underlying and proximate determinants. In this way, the authors could express the analysis in HR instead of OR.

8. Minor issues: It is necessary to add a space between some words and references throughout the text.

Reviewer #2: This is an important study in adolescent and young females at risk of acquiring HIV with a detailed assessment of socioeconomic factors associated.

Few comments:

I would add in the introduction the access of the population studied to preventive tools (testing, attesting, condoms and prep) if that is the case. In the result section I would change the column of baseline characteristics of 15-24 yo to "Total of participants", to make it more clear for readers. The 27% of participants not completing the follow-up, could you provide the characteristics as well in the text? It might be that those have several differences in their social determinants that may have drive them to stop the follow up and I think that is important to address. Regarding the missingness of many variables, I think it would be fair to discuss this more in detail in the discussion section, although it is established as a limitation, I think it deserves more potential explanation to that.

Reviewer #3: Peer Review Template

HIV incidence and associated risk factors in adolescent girls and young women in South Africa: A population-based cohort study

1. Summary of the research

Lewis and colleagues studied socio-demographic, behavioural and biological determinants of HIV incidence among adolescent girls and young women (AGYW) in South Africa. They used a proximate determinants framework to define pathways through which HIV infections could occur in this vulnerable group. The authors concluded that retaining AGYW in school, and having their partners access voluntary medical male circumcision, utilize condoms and be on HIV treatment as prevention all reduce risks of their acquisition of HIV. They also concluded that family support reduces AGYW risky sexual behaviour and HIV acquisition. The authors therefore highlighted the importance of designing and implementing community-based HIV combination prevention programs that effectively address the identified structural, biological and behavioural HIV risk factors in their design.

The research findings are fully consistent with the existing literature and add value to the body of evidence in regard to the causal pathways of HIV acquisition.

The manuscript has a number of strengths. It measures the relationships between both proximate and underlying determinants with HIV incidence and elucidates how underlying determinants influence proximate determinants. It also elucidates causal pathways for the HIV acquisition. Finally, it measures HIV incidence levels in sub-groups of AGYW in this area. One study weakness is that there may be social desirability bias which could have led to underreporting of risky sexual behaviour. The authors however acknowledge this weakness to be taken into account in the interpretation of the study findings.

My overall recommendation is that this is a technically sound manuscript that is well written. It should be accepted with very minor revisions.

2. Examples and evidence

2.1. Major issues

There are no major issues with the manuscript.

2.2. Minor issues

2.2.1. In the introduction, line 62, you could consider specifying to what extent the risk in this sub-group has declined, and to what extent the risk of HIV remains substantial.

2.2.2. In line 65 also in the introduction, you could consider replacing “campaigns” with another word, like “efforts”.

2.2.3. Line 223-225 in the results section refers to AGYW 15-19 years old who received support. The subsequent sentence (line 225-226) gives the incidence among AGYW who did not receive support. Though the incidence among AGYW who received support in included in Table 1, for easier comparison, it could help to include the incidence among those who received support in the text.

2.2.4. For line 229-231 also in the results section, it may be important to report what proportion of AGYW who reported knowingly having sex with individuals who were HIV positive and not on ART did not use protection.

3. Other points (optional)s

None

6. PLOS authors have the option to publish the peer review history of their article (what does this mean?). If published, this will include your full peer review and any attached files.

Reviewer #1: No

Reviewer #2: No

Reviewer #3: **Yes: **Brian C Chirombo, MPH, MBChB

---

## [Author Response · Author response to Decision Letter 0]

2 Sep 2022

1. Thank you for the feedback from the PLOS ONE reviewers on our manuscript entitled “HIV incidence and associated risk factors in adolescent girls and young women in South Africa: A population-based cohort study”. We appreciate the opportunity to revise this manuscript. Below we provide a point-by-point response to each of the reviewer comments, with reference to parts of the manuscript that have been updated. 

2. Reviewer 1 comments

This is a cohort study aiming to identify determinants of HIV incidence in adolescent girls and young women in a hyperendemic setting in South Africa. It is a very interesting study; however, it presents some gaps, especially in the methodology, and must be improved.

2.1 I missed the sample size estimation in the Methods section. Is your sample size statistically significant, and the findings of this study can be generalized?

Response - Thank you for this observation. We have added a reference to the study protocol in the Methods section as follows: “Further details of the study have been previously published [26, 5].” The HIPSS protocol paper outlines the sample size calculations used in the study. In short, the HIPSS study was powered to detect a difference in the HIV incidence in the two study cohorts. The sample size provided 84 % power to detect a 30 % reduction in HIV incidence rate at a 5 % significance level, given an HIV prevalence of 20 %, loss-to-follow-up of 15 % per annum and an initial HIV incidence rate of 3 per 100 person years. This particular study was based on the results from a sub-analysis of HIPSS data and does not look at the change in incidence but rather predictors of incidence. In total there were 163 HIV seroconversions – a considerably high number given the limited age range of the sample – and these events allowed us to perform an extensive analysis of potential predictors of HIV. 

2.2 The inclusion and exclusion criteria are not clearly stated. They should be presented in the Study design and setting section.

Response - Criteria for inclusion into the cohort were based on age, HIV status and willingness to provide blood samples for laboratory testing. Only one person was selected per household and this person was selected at random among the eligible individuals in a household. We have outlined these criteria in the following lines in the Methods section:

“One individual per household was selected at random and enrolled in the survey on condition they were aged between 15 and 49 years and provided peripheral blood samples for laboratory HIV and pregnancy testing.”

“Individuals were enrolled in the cohorts if they were HIV negative at survey enrolment and aged between 15 and 35 years.”

2.3 An important piece of information not available in the paper is how the two cohorts were linked.

Response – Thank you for identifying this important gap. We have amended the Methods section to clarify this by including the following text: “HIPSS comprised of two serial cross-sectional household surveys, with two embedded HIV-negative cohorts comprising of a single follow-up visit. The first survey was conducted between June 2014 and June 2015 and the second between June 2015 and June 2016, and the follow-up visits were completed by January 2017 and August 2017 respectively. Fingerprint biometrics were used to confirm the identity of eligible participants for the follow-up visit. Individuals could be included in both surveys and cohorts if selected, however the overlap was minimal [5].”

2.4 How did you analyze the associated factors by statistical or theoretical criteria? In fact, the authors didn’t focus on how this analysis was done in the Statistical analysis section, although this is one of the paper's objectives.

Response - Factors included in analysis were selected based on their availability in the HIPSS survey and evidence of association in past literature, as stated in the text: “Underlying and proximate determinants comprised socio-demographic, behavioural and biological factors that were measured in the HIPSS survey and ones that have been determined as influencing HIV incidence in literature [20, 27, 28].”

The analysis was structured using a proximate determinants framework as outlined in the section entitled “Conceptual approach”. Cox proportional hazards modelling was then used to model the association between potential risk factors and HIV incidence. We performed a univariable regression followed by a multivariable and we have amended the statistical section to make this clearer: “The association between identified underlying and proximate determinants and HIV incidence was estimated using Cox proportional hazards models. Since data on orphan status and STI testing data was only collected for one of the two cohorts, orphan and STI status were excluded from the Cox regression although included in descriptive analysis. Univariable regression was first performed followed by multivariable regression. All variables included in the univariable were included in the multivariable regression as all variables were hypothesized to be associated with HIV incidence.”

A significance level of 0.05 was used to guide whether associations were considered to be strong or not. The direction of the association – whether it was positive or negative – was determined by whether the hazard ratio was above or below 1.

2.5 Why did you use logistic regression to estimate the association between the underlying and proximate determinants significantly associated with HIV incidence? 

Response - The proximate determinants found to be significantly associated with HIV incidence were: number of partners during follow-up (2 or more vs 1), partner(s) reported HIV and ART status (at least one positive and not on ART vs not), partner circumcision status (all circumcised vs not) and condom use (inconsistent or no condom use vs consistent condom use). As these variables are all binary, a logistic regression was the appropriate choice of model for the analysis when analysing the association between these variables and underlying determinants.

2.6 Role of the Funder/Sponsor: This section can be stated at the end of the paper, after the conclusions.

Response - The PLOS ONE's style requirements (found here: https://journals.plos.org/plosone/s/file?id=wjVg/PLOSOne_formatting_sample_main_body.pdf ) did not indicate that we could put this section after the conclusions section. We have removed this section as it will be covered in the financial disclosure section if successfully published.

2.7 Association between underlying and proximate determinants: This analysis is very confusing! I think a hierarchical analysis model would be better in this context, splitting by underlying and proximate determinants. In this way, the authors could express the analysis in HR instead of OR.

Response - The association between underlying and proximate determinants and HIV incidence was measured using multivariable Cox regression with results reported in Table 3. This analysis looked at the association between a factor and HIV incidence after adjusting for all other measured factors. Following this analysis, we looked at the association between underlying determinants and proximate determinants as presented in table 4. This analysis was designed to: “explore how underlying determinants may influence proximate determinants of HIV incidence”. We hypothesize- based on the proximate determinants framework - that underlying determinants “operate through proximate determinants to influence the likelihood of being exposed to HIV”. Thus, to test this hypothesis we analyse the association between underlying and proximate determinants with the proximate determinants as the outcome variable. As the proximate determinants variables are binary in nature and have no time element, we cannot use a time-to-event analysis like Cox proportional hazards modelling for this analysis. Logistic regression is a suitable chose for analyses with a binary outcome variable. 

2.8 Minor issues: It is necessary to add a space between some words and references throughout the text. 

Response - Thank you, this has been addressed.

 

3. Reviewer 2 comments

This is an important study in adolescent and young females at risk of acquiring HIV with a detailed assessment of socioeconomic factors associated. Few comments:

3.1 I would add in the introduction the access of the population studied to preventive tools (testing, attesting, condoms and prep) if that is the case. 

Response - Thank you. We have added this sentence to the Study design and setting section: “Contraceptive services and HIV testing and treatment are freely available through primary health care clinics and, since 2016, oral pre-exposure prophylaxis (PrEP) has been made available to people at substantial risk of HIV infection.” To illustrate that PrEP usage was low in the cohort, we have included frequency of PrEP usage to table 1.

3.2 In the result section I would change the column of baseline characteristics of 15-24 yo to "Total of participants", to make it more clear for readers. 

Response - Thank you. We have amended this column to have the heading “Total”.

3.3 The 27% of participants not completing the follow-up, could you provide the characteristics as well in the text? It might be that those have several differences in their social determinants that may have drive them to stop the follow up and I think that is important to address. 

Response - Thank you. 23% were not accessible for follow-up. We have added the following text to the results section to provide some insight into this group: “Compared to those with follow-up data, HIV-negative AGYW who were not available for follow-up (n=808, 23.0%) were more likely to be aged between 20 and 24 (48.2% versus 54.7%) and reside in a rural area (51.7% versus 76.6%). However, the two groups were comparable with respect to education levels, household income, number of lifetime sexual partnerships and STI status at baseline.”

3.4 Regarding the missingness of many variables, I think it would be fair to discuss this more in detail in the discussion section, although it is established as a limitation, I think it deserves more potential explanation to that.

Response - Thank you for identifying this. We have added the following text to the discussion section: “Missing data can lead to a loss of statistical power or introduce an unexpected selection bias. However, in this study the number of missing responses to questions in the baseline survey and cohort follow-up visit were generally low, and a comparison of characteristics of those lost to follow-up to those who were followed did not suggest that the two groups were inherently different.”

 

4. Reviewer 3 comments

Lewis and colleagues studied socio-demographic, behavioural and biological determinants of HIV incidence among adolescent girls and young women (AGYW) in South Africa. They used a proximate determinants framework to define pathways through which HIV infections could occur in this vulnerable group. The authors concluded that retaining AGYW in school, and having their partners access voluntary medical male circumcision, utilize condoms and be on HIV treatment as prevention all reduce risks of their acquisition of HIV. They also concluded that family support reduces AGYW risky sexual behaviour and HIV acquisition. The authors therefore highlighted the importance of designing and implementing community-based HIV combination prevention programs that effectively address the identified structural, biological and behavioural HIV risk factors in their design. The research findings are fully consistent with the existing literature and add value to the body of evidence in regard to the causal pathways of HIV acquisition. The manuscript has a number of strengths. It measures the relationships between both proximate and underlying determinants with HIV incidence and elucidates how underlying determinants influence proximate determinants. It also elucidates causal pathways for the HIV acquisition. Finally, it measures HIV incidence levels in sub-groups of AGYW in this area. One study weakness is that there may be social desirability bias which could have led to underreporting of risky sexual behaviour. The authors however acknowledge this weakness to be taken into account in the interpretation of the study findings. My overall recommendation is that this is a technically sound manuscript that is well written. It should be accepted with very minor revisions.

Response - Thank you for reviewing our manuscript and for your encouraging feedback!

4.1 In the introduction, line 62, you could consider specifying to what extent the risk in this sub-group has declined, and to what extent the risk of HIV remains substantial.

Response –While all referenced manuscripts suggest that incidence is declining, the magnitude of the reported declines differs by methodology and location of research. As such, we have decided not to include this detail. The extent of the risk of HIV for AGYW in this region is summarised in the line “approximately 4,200 AGYW became infected with HIV every week in 2020 [2].”

4.2 In line 65 also in the introduction, you could consider replacing “campaigns” with another word, like “efforts”.

Response- We have changed this to ‘programmes’.

4.3 Line 223-225 in the results section refers to AGYW 15-19 years old who received support. The subsequent sentence (line 225-226) gives the incidence among AGYW who did not receive support. Though the incidence among AGYW who received support in included in Table 1, for easier comparison, it could help to include the incidence among those who received support in the text. 

Response - We have added this detail: “Among those that did not receive this support, HIV incidence was 10.40 (95% CI: 5.67-19.09) per 100 person-years compared to 3.17 (95% CI: 2.38-4.21) among those who did.” 

4.4 For line 229-231 also in the results section, it may be important to report what proportion of AGYW who reported knowingly having sex with individuals who were HIV positive and not on ART did not use protection.

Response - Only 5(10%) of the 48 reported consistent condom use in the baseline and follow-up questionnaires. We have not added this detail as it could not be added to Table 1 and hence reference to the result may have confused the reader.

---

## [Decision Letter · Decision Letter 1]

4 Nov 2022

PONE-D-22-19202R1HIV incidence and associated risk factors in adolescent girls and young women in South Africa: A population-based cohort studyPLOS ONE

Dear Dr. Lewis,

Thank you for submitting your manuscript to PLOS ONE. After careful consideration, we feel that it has merit but does not fully meet PLOS ONE’s publication criteria as it currently stands. Therefore, we invite you to submit a revised version of the manuscript that addresses the points raised during the review process.

We look forward to receiving your revised manuscript.

Kind regards,

Hamid Sharifi

Academic Editor

PLOS ONE

Journal Requirements:

Reviewers' comments:

Reviewer's Responses to Questions

**Comments to the Author**

1. If the authors have adequately addressed your comments raised in a previous round of review and you feel that this manuscript is now acceptable for publication, you may indicate that here to bypass the “Comments to the Author” section, enter your conflict of interest statement in the “Confidential to Editor” section, and submit your "Accept" recommendation.

Reviewer #1: All comments have been addressed

Reviewer #2: (No Response)

Reviewer #3: All comments have been addressed

2. Is the manuscript technically sound, and do the data support the conclusions?

Reviewer #1: Yes

Reviewer #2: Yes

Reviewer #3: Yes

3. Has the statistical analysis been performed appropriately and rigorously? 

Reviewer #1: Yes

Reviewer #2: Yes

Reviewer #3: Yes

4. Have the authors made all data underlying the findings in their manuscript fully available?

Reviewer #1: Yes

Reviewer #2: Yes

Reviewer #3: Yes

5. Is the manuscript presented in an intelligible fashion and written in standard English?

Reviewer #1: Yes

Reviewer #2: Yes

Reviewer #3: Yes

6. Review Comments to the Author

Reviewer #1: Thank you for answering my questions and adding my suggestions. I congratulate the authors for the manuscript.

Reviewer #2: This is an important Study in a specific population (female adolescents) at high risk of acquiring HIV and their findings could help to design future interventions.

Some comments and editions:

Abstract: In results authors do not describe the overall results, they only describe the risk factors in 2 groups: those 15-19 and 20-24 years. It should be describen first the overall results and then perhaps divided according to range of ages. Additionally, division of ranges of ages, are not pre-specified in the methods sections. Authors, should describe all findings in the group of 15-19 years, and then all the results of population of 20-24 years in order to read it better.

Intro: I could suggest to add something about the fact that adolescent population have proven in clinical trials to be a very challenging population to enroll, adhere and retain in treatment and prevention of HIV... therefore, understand the characteristics of those acquiring HIV would help to design better interventions.

Methods: Here are my main concerns about this study... What if through the multistage sampling (only one household enrolled) you are missing those who are at most risk...? Additionally, only 77% completed. I would analyze the population that completed vs those who did not, in order to show if those are different population. If they are, then results should be interpret with more caution. On the other hand, is there a national registry in which you could compare your incident cases with those registered... It might be that the incidence is different from the reality, just because the sampling could miss those more at risk...?

Additionally, in methods you should pre specify the way you are going to analyze and stratify your population. In results you present different groups: 15-19 years, 20-24 years, then the analysis of those with less education level, also those without sexual history, etc. All groups should be described in methods and the rationale of those divisions.

Results: Comparison with their peers male is presented, not explained in methods how this population was analyzed or picked... It is a very interesting comparison, but it has to be explained in methods in order to see that the comparison is fair and reliable.

Finally, the data were collected from 2014-2017... prevention programs have changed in the last decade. I would like to know why is the data so old and discuss this issue in the discussion section and as a limitation.

Reviewer #3: (No Response)

7. PLOS authors have the option to publish the peer review history of their article (what does this mean?). If published, this will include your full peer review and any attached files.

Reviewer #1: No

Reviewer #2: No

Reviewer #3: **Yes: **Brian C Chirombo, MBChB, MPH

---

## [Author Response · Author response to Decision Letter 1]

16 Nov 2022

1. Journal Requirements:

We have reviewed all references and updated accordingly.

2. Reviewer 2 comments

This is an important Study in a specific population (female adolescents) at high risk of acquiring HIV and their findings could help to design future interventions.

Some comments and editions:

Abstract: In results authors do not describe the overall results, they only describe the risk factors in 2 groups: those 15-19 and 20-24 years. It should be describen first the overall results and then perhaps divided according to range of ages. Additionally, division of ranges of ages, are not pre-specified in the methods sections. Authors, should describe all findings in the group of 15-19 years, and then all the results of population of 20-24 years in order to read it better.

response: Thank you. We have re-worded the results so that the 15-19-related results appear first followed by those for the 20-24 year-olds. We have added to the methods that separate models were built for the two age groups. In the body of the manuscript, we explain why we took the approach to split the age groups: “Analyses were performed separately for 15-19-year-olds and 20-24-year olds as we hypothesized that the factors affecting young girls who are still of school going age are likely to be different from those affecting women who are out of secondary-school, possibly in tertiary education or seeking employment.” We provide overall incidence results in the abstract, but since separate models were used for measuring the associations for the two age groups, we cannot combine these results to one overall estimate. 

Intro: I could suggest to add something about the fact that adolescent population have proven in clinical trials to be a very challenging population to enroll, adhere and retain in treatment and prevention of HIV... therefore, understand the characteristics of those acquiring HIV would help to design better interventions. Methods: Here are my main concerns about this study... What if through the multistage sampling (only one household enrolled) you are missing those who are at most risk...? Additionally, only 77% completed. I would analyze the population that completed vs those who did not, in order to show if those are different population. If they are, then results should be interpret with more caution. On the other hand, is there a national registry in which you could compare your incident cases with those registered... It might be that the incidence is different from the reality, just because the sampling could miss those more at risk...?

response: Thank you for raising this important concern. The survey enrolled one randomly selected person per household (not one household). Our incidence is higher than that reported based on the national estimates for 15-24 year olds (South African National HIV Prevalence, Incidence, Behaviour and Communication Survey, 2017). This is expected given the high HIV prevalence in the region in which the survey was conducted but also provides us with reassurance that our selection – while random – were not unduly biased by refusal rates. The original version of this manuscript was updated with this text in the manuscript: “Compared to those with follow-up data, HIV-negative AGYW who were not available for follow-up (n=808, 23.0%) were more likely to be aged between 20 and 24 (48.2% versus 54.7%) and reside in a rural area (51.7% versus 76.6%). However, the two groups were comparable with respect to education levels, household income, number of lifetime sexual partnerships and STI status at baseline.”

Additionally, in methods you should pre specify the way you are going to analyze and stratify your population. In results you present different groups: 15-19 years, 20-24 years, then the analysis of those with less education level, also those without sexual history, etc. All groups should be described in methods and the rationale of those divisions.

response: The reason for splitting the age groups is provided in the methods: “Analyses were performed separately for 15-19-year-olds and 20-24-year olds as we hypothesized that the factors affecting young girls who are still of school going age are likely to be different from those affecting women who are out of secondary-school, possibly in tertiary education or seeking employment.” 

We outline where we exclude those without sexual history and why here in the methods: “Unless otherwise stated, all AGYW, regardless of whether they reported having sex before study enrolment, were included in the analysis. However, models that incorporated sexual behaviour variables that were based on follow-up data excluded, by necessity, AGYW who reported not being sexually active in the 12 months preceding follow-up.”

The analysis of the association between education and other sexual behavioral variables is explained here in the methods: “The association between the underlying determinants and proximate determinants found to be significantly associated with HIV incidence was measured using logistic regression.”

Results: Comparison with their peers male is presented, not explained in methods how this population was analyzed or picked... It is a very interesting comparison, but it has to be explained in methods in order to see that the comparison is fair and reliable.

response: We are uncertain which part of the manuscript this comment relates to. We think this comment relates to an earlier version of the manuscript where an incidence of 0.82 was reported for male peers. This line was removed in the first revision of the manuscript for the reasons the reviewer provides.

Finally, the data were collected from 2014-2017... prevention programs have changed in the last decade. I would like to know why is the data so old and discuss this issue in the discussion section and as a limitation.

response: The efforts of the authors writing the manuscript were redirected to COVID-19 work during 2020 and 2021. We have now highlighted the age of the data in the limitations section. While HIV prevention programmes have changed in the last decade, determinants of risk in young women appear to remain quite consistent – education, family support age-disparate partnerships and high STI prevalence repeatedly present themselves as related factors.

---

## [Decision Letter · Decision Letter 2]

5 Dec 2022

HIV incidence and associated risk factors in adolescent girls and young women in South Africa: A population-based cohort study

PONE-D-22-19202R2

Dear Dr. Lewis,

We’re pleased to inform you that your manuscript has been judged scientifically suitable for publication and will be formally accepted for publication once it meets all outstanding technical requirements.

Kind regards,

Hamid Sharifi

Academic Editor

PLOS ONE

Additional Editor Comments (optional):

Reviewers' comments:

Reviewer's Responses to Questions

**Comments to the Author**

1. If the authors have adequately addressed your comments raised in a previous round of review and you feel that this manuscript is now acceptable for publication, you may indicate that here to bypass the “Comments to the Author” section, enter your conflict of interest statement in the “Confidential to Editor” section, and submit your "Accept" recommendation.

Reviewer #2: All comments have been addressed

2. Is the manuscript technically sound, and do the data support the conclusions?

Reviewer #2: Yes

3. Has the statistical analysis been performed appropriately and rigorously? 

Reviewer #2: Yes

4. Have the authors made all data underlying the findings in their manuscript fully available?

Reviewer #2: Yes

5. Is the manuscript presented in an intelligible fashion and written in standard English?

Reviewer #2: Yes

6. Review Comments to the Author

Reviewer #2: Authors have nicely addressed all my queries, questions and suggestions. I think this is a very nice manuscript with important data, ready for publication.

Thank you for considering me as reviewer.

7. PLOS authors have the option to publish the peer review history of their article (what does this mean?). If published, this will include your full peer review and any attached files.

Reviewer #2: **Yes: **Brenda Crabtree-Ramirez

---

## [Editor Report · Acceptance letter]

13 Dec 2022

PONE-D-22-19202R2 

HIV incidence and associated risk factors in adolescent girls and young women in South Africa: A population-based cohort study 

Dear Dr. Lewis:

I'm pleased to inform you that your manuscript has been deemed suitable for publication in PLOS ONE. Congratulations! Your manuscript is now with our production department. 

Kind regards, 

on behalf of

Dr. Hamid Sharifi 

Academic Editor

PLOS ONE